# 2N+4-rule and an atlas of bulk optical resonances of zigzag graphene nanoribbons

Renebeth B. Payod [1], Davide Grassano [2], Gil Nonato C. Santos[1], Dmitry I. Levshov[3,6], Olivia Pulci[2] & Vasil A. Saroka [4,5]*

Development of on-chip integrated carbon-based optoelectronic nanocircuits requires fast and non-invasive structural characterization of their building blocks. Recent advances in synthesis of single wall carbon nanotubes and graphene nanoribbons allow for their use as atomically precise building blocks. However, while cataloged experimental data are available for the structural characterization of carbon nanotubes, such an atlas is absent for graphene nanoribbons. Here we theoretically investigate the optical absorption resonances of armchair carbon nanotubes and zigzag graphene nanoribbons continuously spanning the tube (ribbon) transverse sizes from 0.5(0.4) nm to 8.1(12.8) nm. We show that the linear mapping is guaranteed between the tube and ribbon bulk resonance when the number of atoms in the tube unit cell is $2N + 4$, where $N$ is the number of atoms in the ribbon unit cell. Thus, an atlas of carbon nanotubes optical transitions can be mapped to an atlas of zigzag graphene nanoribbons.

[1] Physics Department, De La Salle University, 2401 Taft Avenue, 0922 Manila, Philippines. [2] Department of Physics, and INFN, University of Rome Tor Vergata, Via della Ricerca Scientifica 1, Rome I-00133, Italy. [3] Faculty of Physics, Southern Federal University, 5 Zorge Str., Rostov-on-Don 344090, Russia. [4] Center for Quantum Spintronics, Department of Physics, Norwegian University of Science and Technology, NO-7491 Trondheim, Norway. [5] Institute for Nuclear Problems, Belarusian State University, Bobruiskaya 11, 220030 Minsk, Belarus. [6] Present address: Physics Department, University of Antwerp, Universiteitsplein 1, B-2610 Antwerp, Belgium. *email: vasil.saroka@ntnu.no

Single-walled carbon nanotubes (SWCNT) and monolayer graphene nanoribbons (GNR) are quasi-one-dimensional nanostructures of graphene that are considered as key building blocks for on-chip integrated carbon optoelectronics[1–3]. The ultimate goal of on-chip integration requires a deep understanding of the structure-function interrelations and a perfect geometrical control of the building blocks. Significant advancements in the synthesis of graphene ribbons with atomic precision have been observed in recent years[4,5], whereas a comprehensive progress has been achieved in production of monochiral carbon nanotubes[2,6,7]. This development has partially eliminated both the edge quality issue of graphene nanoribbons and the chirality problem for carbon nanotubes. At the same time, considerable effort has been directed towards relating SWCNT optical properties for different polarization of radiation with SWCNT chirality and diameter[8–16].

A standard way of presenting diameter dependence of SWCNT optical resonances polarized parallel to the tube axis is Kataura plot[8]. This plot has been both measured experimentally[9,17] and reproduced numerically in semi-empirical and ab-initio calculations[18–20]. Several interpolating formulas have been proposed for the description of the SWCNT Kataura plot[10–12]. The Kataura plot for ribbons, however, cannot be approached in the same way as Kataura plot for tubes because of two main reasons: (i) the different physical mechanisms of SWCNT and GNR synthesis and (ii) a huge variety of GNR edges, which impedes the development of the GNR standard classification, thereby leading to different naming conventions used in experimental and theoretical studies[21–23]. Limiting our consideration to the symmetric nanoribbons with zigzag and armchair edges, we should notice that even these two types of GNRs are not yet well studied and understood. In particular, no Kataura plot has been reported for these graphene nanoribbons, although such plot would be practical for the fast and non-invasive ribbon width characterization, especially in those synthesis methods that provide flexible control of the ribbon width, such as catalytic writing, nanolithography, plasma etching, tube unzipping or epitaxial growth on sidewalls of SiC[24–26].

The optical properties of armchair GNRs (AGNRs) have been intensively studied theoretically[27–36] and more recently experimentally[37,38], nevertheless the absorption peak dependence on the ribbon width is not yet complete. Hitherto, rather narrow energy or width ranges have been reported. The width dependence of the first ($E_{11}$) and the second ($E_{22}$) single-electron excitation was exemplified for 0.3–3 nm widths[29] while the exciton $E_{11}$ was studied in the AGNRs of 0.4–4 nm widths[32]. The $E_{22}$ and $E_{33}$ excitons were also reported for the similar width ranges[33–35]. Recently, an analytical width dependence has been proposed for $E_{11}$ exciton of semiconducting AGNRs[36]. Even so, none of these previous works have provided a proper comparison of the AGNR optical resonances with those of zigzag SWCNTs, whereas such comparative analysis is highly desirable for a reliable optical nanodevice and nanocircuit engineering. Joint analysis of several previous studies attempting to compare zigzag SWCNTs and AGNRs indicates that Kataura plots for these ribbons could be the replica of that plot for zigzag SWCNTs. As shown within the tight-binding model, the electronic energy bands of zigzag SWCNT($n$,0) duplicate the bands of the AGNR($w$) when $n = w + 1$, where $n$ is the chiral or diameter index of the tube and $w$ is the width index of the ribbon[39,40]. Similar correspondence takes place for the optical transition matrix elements[41,42], leading to a perfect alignment of the single-electron optical absorption peaks of some SWCNTs and corresponding AGNRs[40]. Moreover, this alignment should not be disordered by excitonic effects since Loudon's model provide the same exciton binding energy for two structures of the same

transverse size[43]. Thus, Kataura plot for AGNRs could, in principle, replicate that plot for zigzag SWCNTs.

The situation is more complicated with zigzag graphene nanoribbons (ZGNRs). These nanoribbons are famous for their edge states[44–46], whose origin is traced back to polyacetylene[47,48]. Optical properties of ZGNRs have been studied theoretically within the first principles and tight-binding models[27,28,30,40,49–52] and experimentally with some edge modifications[53]. However, the majority of these studies are focused on the effect of the edge states and on few selected ribbons. Only few works attempt to consider a wide range of ZGNRs. The lowest five optical transitions have been reported for wide ribbons of 12.8–76.7 nm[30]. This study, however, omits the relation of these wide ribbon optical transitions to the transitions of armchair SWCNTs. The width dependence for the first four optical transitions is presented for narrow ZGNRs of 0.5–4.5 nm in ref. [49]. These results are compared to the transition energies of armchair SWCNTs through the ratio $E_{11} : E_{22} : E_{33} : E_{44}$. However, the calculated optical resonance energies in ref. [49] are the energy differences between the van Hove singularities symmetrically placed with respect to the intrinsic Fermi level. Associating these energies with the optical resonances contradicts to the optical selection rules[30,31,52,54]. Not only are the selection rules different for ZGNRs and armchair SWCNTs but also their energy bands cannot be aligned. Nevertheless, a hidden correlation between the optical absorption resonances of several ZGNRs and armchair SWCNTs has been reported in the nearest-neighbor tight-binding model[40,52]. Therefore, ZGNRs and armchair SWCNTs deserve primary consideration to provide an entry point for a general understanding of the optical interrelations between SWCNTs and GNRs.

In this paper, we study the interrelations between optical absorption resonances in ZGNRs and armchair SWCNTs for the parallel polarization of the incident light. Accounting for the exchange and correlation effects, we analyze the degree of linear correlation and alignment between the optical resonances as a function of the tube diameter and ribbon width. We reveal a $2N + 4$-rule according to which the maximum of the linear correlation and alignment is observed when the effective width of the ribbon is equal to the half of the tube circumference, which is equivalent to the number of carbon atoms in the armchair SWCNT unit cell being $2N + 4$, where $N$ is the number of carbon atoms in a zigzag GNR. We apply this rule to the atlas of SWCNTs[11] and predict the resonance energies for the bulk-bulk transitions of ZGNRs.

## Results

**Optical absorption of armchair SWCNTs and zigzag GNRs.** In this letter, we use the integers $n$ and $w$ as the tube diameter and ribbon width indices, respectively[40]. These indices can be readily converted to the actual diameter of the tube $d$ and width of the ribbon $W$ as follows: $d = \sqrt{3}an/\pi$ and $W = \sqrt{3}aw/2$, with $a = 2.46$ Å being the graphene lattice constant. The intensity of the absorption as a function of the structure transverse size and excitation energy is presented in Fig. 1 (see Methods for details). These graphs are an analog of the celebrated Kataura plots[8,9] for SWCNTs with a difference that these diagrams only present the absorption resonances for the ZGNRs (Fig. 1a, b) and armchair SWCNTs (Fig. 1c, d). From multiple SWCNT experimental studies[9–12], the first four[10,12] and even higher[11] absorption peaks can be recovered by the interpolating formulas. However, no experimental data and interpolating formulas are available for the zigzag GNRs. Comparing the results in Fig. 1a, c obtained for the tubes and ribbons in the nearest-neighbor Partoens 2006[55] tight-binding model (TBM), one can clearly identify two regions in the

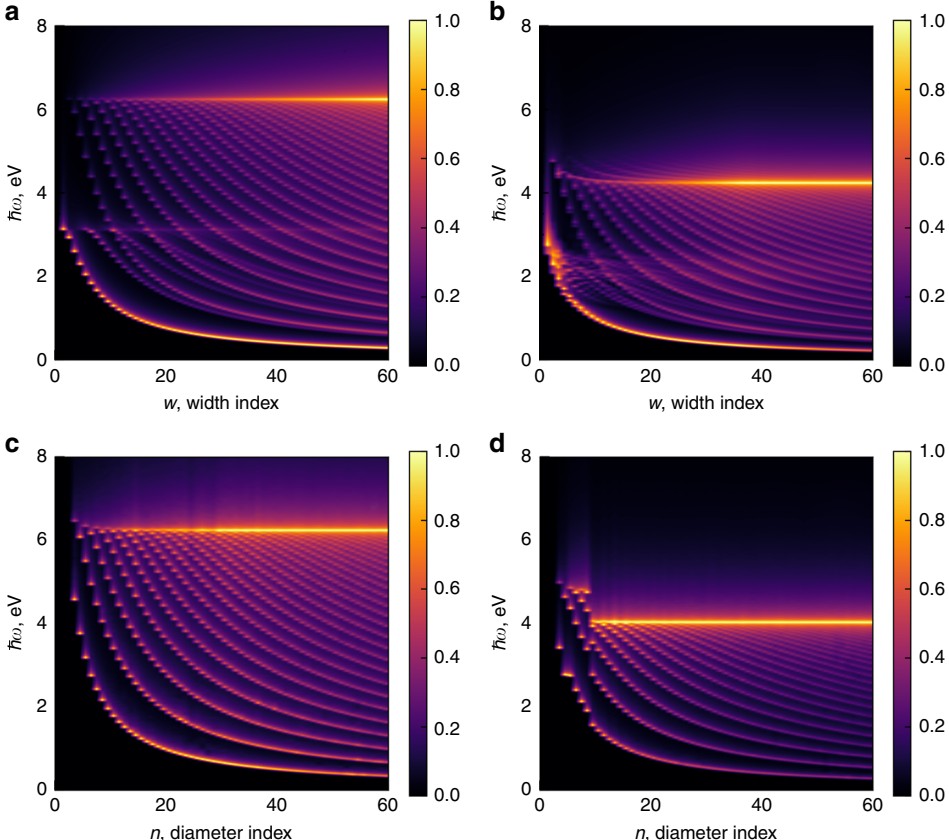

**Fig. 1 Absorption intensity density plots.** Optical absorption dependence on the excitation energy and transverse size of ZGNRs (**a**, **b**) and armchair SWCNTs (**c**, **d**) in various tight-binding models: **a**, **c** the nearest-neighbor tight-binding model (Partoens 2006)[55] **b** model fitted to Quantum Espresso DFT data (ZGNRs (av.)), and **d** model fitted to SIESTA DFT data (Reich 2002)[60].

absorption spectra of the ribbons. The first region at $\hbar\omega \gtrsim 3$ eV has a strikingly similar pattern to the intensity plot of the tubes, while the second region at $\hbar\omega \lesssim 3$ eV contains additional peaks converging to the threshold of $t_1 \approx 3$ eV as the ribbon width decreases. These additional resonances originate from the edge states (see Supplementary Fig. 3 in Supplementary Note 2), and they are noticeably weaker in intensity with the exception of the lowest energy resonance, which is comparable in intensity to the strongest resonance situated at $2t_1 \approx 6$ eV. The high energy resonance at $2t_1 \approx 6$ eV sets the upper boundary for the diverse peak structure in absorption of both tubes and ribbons. It represents the persistent $\pi$-plasmon that is routinely observed in all graphitic materials[56]. By analogy, the much weaker edge $\pi$-plasmon at $t_1 \approx 3$ eV sets the upper boundary for the edge resonances in ZGNRs (see also Supplementary Fig. 3 in Supplementary Note 2). The intensity plots show that all the intensity curves, which split from the 6 eV in the ZGNR, i.e., those which correspond to the bulk-to-bulk state transitions, have counterparts in the intensity plot of the armchair SWCNTs. However, the lowest energy absorption resonances of the armchair SWCNTs do not have a counterpart among the bulk–bulk resonances in the ZGNR absorption spectra.

Advancing to the multi-parameter TBM—with parameters fitted to the Density Functional Theory (DFT) and given in Table 1—we notice in Fig. 1b, d that the main features of the single-parameter model still persist. The boundary between the pure bulk and the bulk-edge transition regions in the ribbons is distinguishable for narrow ribbons but gradually diffuses for wider ones. The $\pi$-plasmon resonance is clearly identified with a red shift to ∼4.0 eV for both the armchair SWCNTs and ZGNRs

(see also Supplementary Figs. 4 and 6). When this resonance is observed for graphene[57–59], the shift between the experimental and theoretical data is typically attributed to the excitonic effects[57,58] or the excitonic effect with Keldysh dielectric confinement and an influence of the substrate[59]. The energy of the $\pi$-plasmon resonance in our calculations is in good agreement with the interpolating formula from ref. [11] (see Supplementary Fig. 8c in Supplementary Note 3), which provides an evidence that indeed the multi-parameter models used here have partially incorporated the many-body effects such as Coulomb electron-electron interactions. In contrast to the diffused edge-bulk transitions resonances found at ∼2 eV in Fig. 1b, the bulk-bulk resonances are less affected by the stated many-body effects. Although the bulk–bulk resonances are squeezed within the range of 0–4 eV due to the many-body effects, the peaks preserve a pattern that exhibits the same number of curves for both SWCNTs and ZGNRs when $n \approx w$. This implies that the energies of the optical absorption resonances in armchair SWCNTs and ZGNRs can be correlated and aligned. These alignment patterns and correlation signature motivate us to study this certain feature in more detail.

**Linear correlation coefficient analysis.** In order to scrutinize a full picture of the existing correlations between the energies of optical resonances in the tubes and ribbons, we have calculated the linear correlation coefficient (LCC) for all possible pairs of SWCNTs and ZGNRs with $n = 4, 5, \ldots, 60$ and $w = 2, 3, \ldots, 60$ (see Methods for details). Two types of LCC are of interest: (i) the LCC between the full set of absorption resonances in ZGNR, including edge-to-bulk transitions, and in armchair SWCNT and

**Table 1 Tight-binding parameters for ZGNRs and armchair SWCNTs.**

|              | $t_0$, eV | $t_1$, eV | $t_2$, eV | $t_3$, eV | $s_1$ | $s_2$ | $s_3$ |
|--------------|-----------|-----------|-----------|-----------|-------|-------|-------|
| ZGNR(6)      | 0.0000    | −2.7163   | −0.0092   | −0.2445   | 0.0008 | 0.0239 | 0.0184 |
| ZGNR(9)      | 0.0000    | −2.7457   | 0.0021    | −0.2331   | 0.0032 | 0.0177 | 0.0222 |
| ZGNR(12)     | 0.0000    | −2.7824   | −0.0080   | −0.2480   | 0.0116 | 0.0131 | 0.0365 |
| ZGNR(15)     | 0.0000    | −2.8020   | 0.0043    | −0.2494   | 0.0134 | 0.0144 | 0.0306 |
| ZGNRs (av.)  | 0.0000    | −2.7616   | −0.0027   | −0.2438   | 0.0072 | 0.0173 | 0.0269 |
| Reich2002[a] | −2.0300   | −2.7900   | −0.6800   | −0.3000   | 0.3000 | 0.0460 | 0.0390 |

[a]Parameters for armchair SWCNTs adapted from ref. [60]

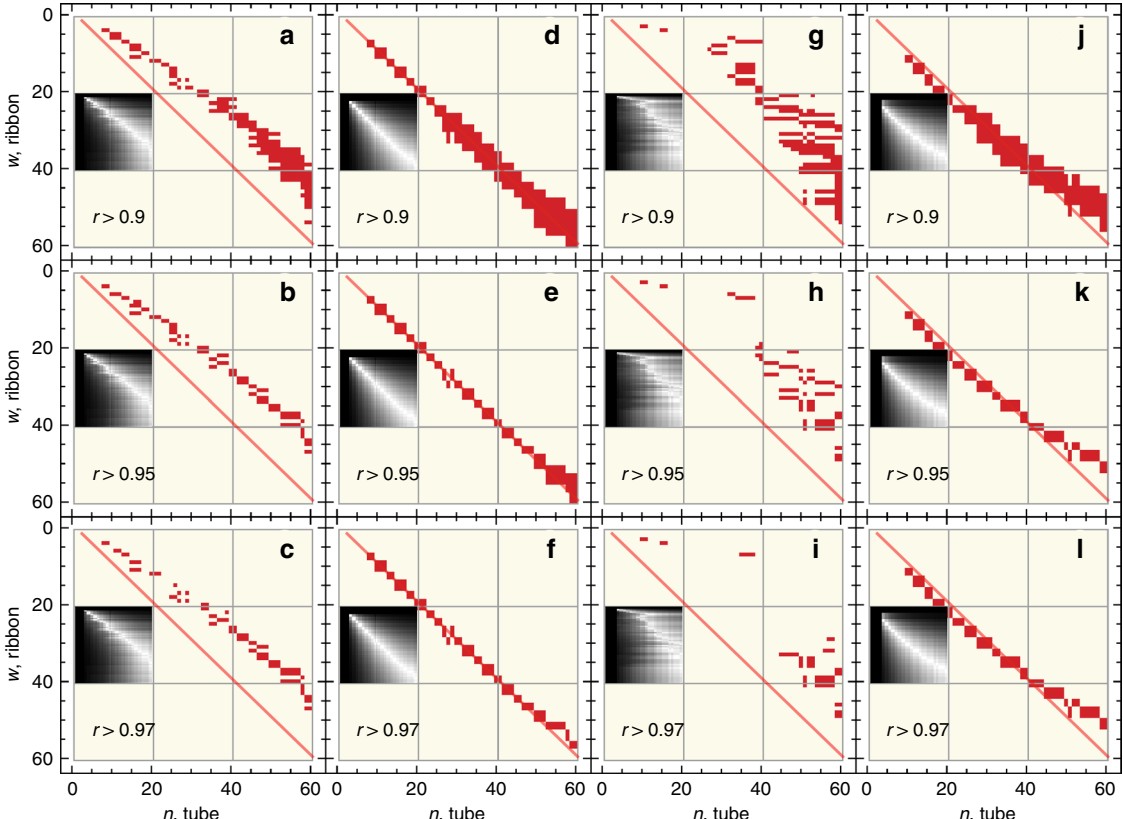

**Fig. 2 Optical resonance linear correlation.** LCC maps for the optical resonances of armchair SWCNTs and ZGNRs: **a–f** single-parameter Partoens 2006 TBM and **g–l** fitted TBMs from Table 1. **a–c** and **g–i** LCC maps include the edge-to-bulk state transitions in ZGNRs. **d–f** and **j–l** LCC maps include only bulk-bulk state transitions in ZGNRs. Insets show full indiscriminate LCC maps, where the color gradient from black to white corresponds to the ascending LCC values. The red regions signify $r > r_{th}$, where $r_{th}$ is the threshold value for the linear correlation coefficient. Red semi-transparent lines are the reference lines given by $n = w + 1$. The lowest energy peak is excluded from the armchair SWCNTs datasets in all cases.

(ii) the LCC between bulk-bulk ZGNR resonances and armchair SWCNTs resonances. The type (i) and (ii) datasets cannot be obtained by analysis of the full absorption spectra of ZGNRs with the peak selection algorithm presented in Methods because it is not selective with respect to the edge-to-bulk or bulk–bulk nature of the optical resonances. An effective approach to isolating such datasets is to apply the peak selection algorithm to the bulk absorption spectra obtained by exclusion of the edge states from the summation in the optical absorption (see Eq. (2) in Methods). As one can clearly see from Supplementary Figs. 3 and 4 in Supplementary Note 2, the inclusion or exclusion of the edge states from the summation does not affect the energies of the bulk-bulk resonances. Therefore, this procedure is equivalent to the selection of the bulk absorption resonances from the full absorption spectra. In both datasets (i) and (ii), the lowest energy resonance of the armchair SWCNT spectra is excluded being the

only tube resonance with no counterpart bulk resonance in the ZGNR absorption spectra (see Supplementary Figs. 5 and 6 in Supplementary Note 2). In Fig. 2, the LCC maps at the threshold $r > r_{th}$ for the single (Fig. 2a–f) and multi (Fig. 2g–l) parameter TBMs are compared with the theoretical reference line $n = w + 1$ obtained by matching the transverse momenta of tube and ribbon electrons[52]. Such matching is achieved by equating the transverse momenta $\theta$ quantized in the two nanostructures by the secular equations: $\cos(n\theta/2) = 1$ (for tube) and $\sin(w\theta) + 2\cos(k/2)\sin[(w + 1)\theta] = 0$ (for ribbon). We note that ZGNR secular equation depends on the electron longitudinal momentum $k$. By setting $k \to 0$, which corresponds to the approximate alignment of the ribbon energy bands to the tube ones at the center of the Brillouin zone[61], the secular equation for the ribbon has solution $\theta_{r,j} = \pi j/(w + 1)$, while the solution for the tube is $\theta_{t,j} = \pi j/n$. The reference line is derived by setting $\theta_{r,j} = \theta_{t,j}$.

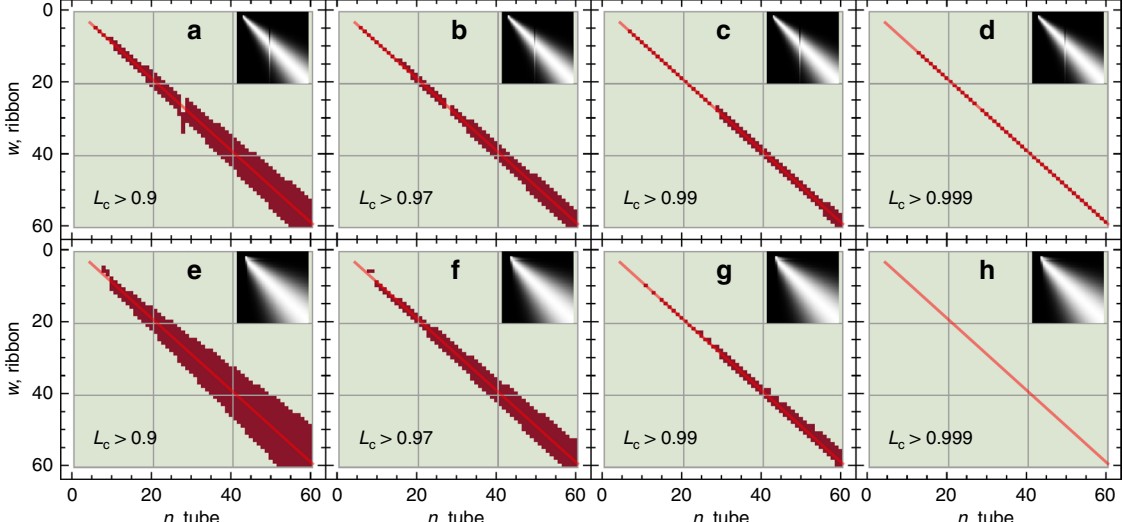

**Fig. 3 Optical resonance alignment.** AC maps for the optical absorption resonances of armchair SWCNTs and the bulk-bulk optical resonances of ZGNRs with **a-d** nearest-neighbor TBM $t_1 = 3.12$ eV[55] and **e-h** fitted TBMs from Table 1. The red line is the reference line $n = w + 1$. Insets of the graphs display the full AC maps before discrimination, where the color gradient from black to white corresponds to the increase of the AC values. The dark red regions signify $L_c > L_{c,th}$, where $L_{c,th}$ is a chosen threshold for the alignment coefficient. The first resonance is excluded from the armchair SWCNTs datasets in all cases.

Allowing the edge-to-bulk state transitions in Partoens2006 TBM and including of the corresponding resonances into the datasets show a significant deviation from the reference line as observed from Fig. 2a–c. However, the exclusion of the edge state optical resonances from the datasets leads to a perfect agreement with the reference line shown in Fig 2d–f. It follows from Fig. 2g–i that in the LCC maps obtained from the multi-parameter TBMs, the high LCC regions form irregular sparse patterns above the reference line if the edge states are involved in the optical absorption. However, a good agreement with the reference line is restored by retaining only the bulk–bulk optical transitions. In summary, the LCC maps in Fig. 2 confirm that the linear relation is warranted for the two-dimensional dataset of optical resonance energies of the armchair SWCNT$(n, n)$ and ZGNR$(w)$ pair when $n = w + 1$. Also, a more detailed comparison between the optical absorption spectra of the selected tube-ribbon pairs (see Supplementary Figs. 5 and 6 in Supplementary Note 2) suggests that the optical resonances are not only linearly related but also precisely aligned.

**Alignment coefficient analysis.** We present in Fig. 3 the maps of the alignment coefficient (AC) $L_c$ discriminated at several thresholds $L_c > L_{c,th}$ and compared with the reference line $n = w + 1$ (see Methods). The regions of high AC obtained from the single-parameter TBM shown in Fig. 3a–d are narrower than those of the multi-parameter TBMs shown in Fig. 3e–h. The broadening of the high AC region in Fig. 3e–h is explained by the squeezing of the absorption resonances presented in Fig. 1. In both types of TBMs, the maximum degree of alignment is observed in the tube-ribbon pairs $n = w + 1$ between the bulk ZGNR and armchair SWCNT resonances. Comparing the panels c–d and g–h of Fig. 3, it can be observed that the degree of alignment in the fitted TBMs is slightly less than that in the nearest-neighbor TBM. Nevertheless, in both cases the AC value exceeds 0.99, which corresponds to the sample variance $\sigma = -\beta \ln(L_c) = 0.01$ eV$^2$.

**2N+4-rule of armchair SWCNTs and ZGNRs.** The high degree of linear correlation and alignment between the optical

resonances in the fitted TBMs presented in Figs. 2 and 3 implies that similar results should be observed for the absorption spectra calculated with the DFT wavefunctions. It can be seen in Fig. 4a, b that similar to ref. [62] the DFT and fitted TBM absorption spectra are in full agreement. It is also seen in Fig. 4a, b that bulk spectra of ZGNRs can be obtained from full spectra by exclusion of the edge states without affecting positions of the bulk resonances. The match between the DFT and TBM absorption spectra for SWCNTs is better than that between the spectra for ZGNRs due to the slight deviation of the TBM energy bands from the DFT ones (see Supplementary Fig. 1 in Supplementary Note 1). Also, Fig. 4a, b indicates that the LCC and AC calculated for the DFT absorption spectra are larger than the corresponding coefficients for the fitted TBMs absorption spectra. Our calculations for the points presented in Fig. 4c show that for the DFT data LCC $r = 0.99987$ and AC $L_c = 0.99934$, while for the fitted TBM data we have LCC $r = 0.83365$ and AC $L_c = 0.98644$.

The correlation and alignment found for the tubes and ribbons mean that the $n = w + 1$ tube-ribbon pairs represent a rule of geometrical matching for the optical resonance alignment. As illustrated in Fig. 4d, the number of carbon atoms in the tube unit cell for these specific pairs is expressed via the number of atoms in the ribbon unit cell $N$ as $2N + 4$.

**An atlas of ZGNR absorption resonances by linear mapping.** The revealed regularity can be employed to predict experimental optical absorption resonances for zigzag GNRs, which have not been explored experimentally yet. We present in Fig. 5 the linear mapping between the optical absorption resonances of armchair SWCNTs and ZGNRs carried out with the parameters extracted from the fitted TBMs in Table 1 (see Supplementary Figs. 7 and 8 in Supplementary Note 3). The data points for ZGNRs presented by the large red open circles in Fig. 5 are described by the following equation:

$$E_{ZGNR} = \left(\frac{\xi_{R,1}}{n} + \zeta_{R,2}\right) E_{aSWCNT} + \left(\frac{\xi_{R,3}}{n} + \zeta_{R,4}\right) \quad (1)$$

where $\xi_{R,1} = 2.83$, $\zeta_{R,2} = 0.98$, $\xi_{R,3} = -7.42$ eV, $\zeta_{R,4} = 0.03$ eV and $E_{aSWCNT}$ is given by the interpolating formula from ref. [11].

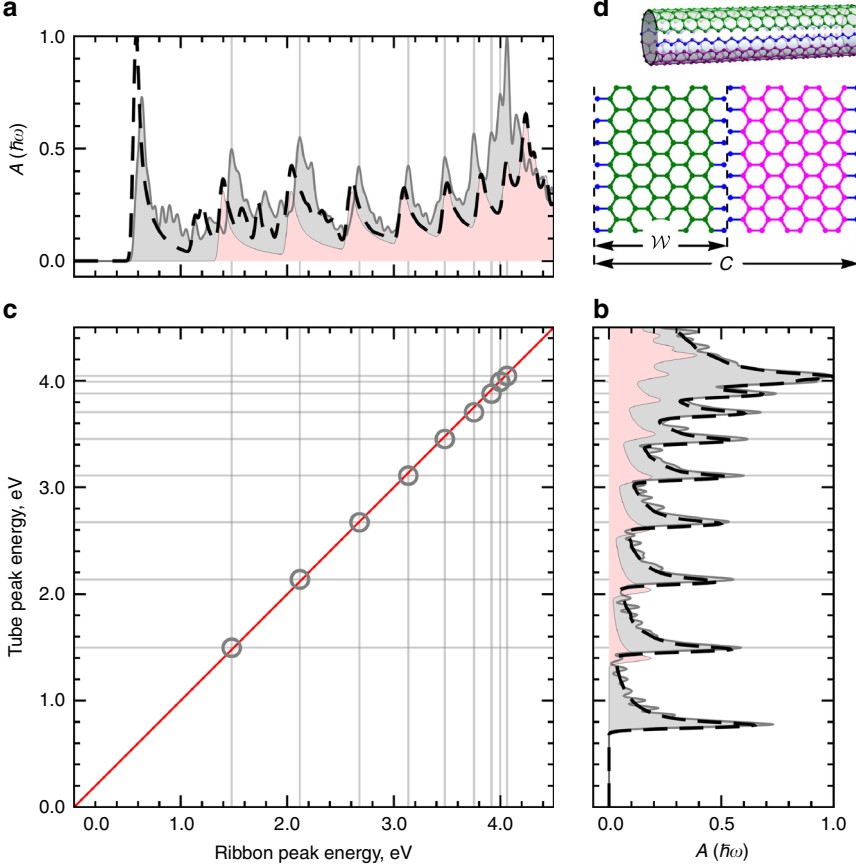

**Fig. 4** $2N + 4$**-rule for optical resonance alignment.** The optical absorption of **a** ZGNR(20) and **b** SWCNT(21, 21) in the DFT (light gray filled) and fitted TBM (black dashed). The light red filling in both **a** and **b** panels represents the bulk optical absorption of ZGNR(20) obtained in the fitted TBM. **c** Two-dimensional energy plot of the ZGNR(20) and SWCNT(21, 21) resonances. Thin gray grid lines in **a**, **b**, and **c** mark the positions of the interband optical resonances extracted from the DFT optical spectra. Gray circles in **c** denote the intersections of the grid lines of most aligned resonance peak energy pairs $(E_{\text{ribbon}}, E_{\text{tube}})$. The red solid line represents a perfect resonance alignment $E_{\text{tube}} = E_{\text{ribbon}}$ used as a reference. **d** Decomposition of SWCNT(7, 7) with circumference $C$ into two ZGNR(6) (magenta and green) with effective width $\mathcal{W}$, which accounts for the two zigzag chains of carbon atoms (blue) to be removed. The effective ribbon width $\mathcal{W} = W + \sqrt{3}a/2$, where $W = \sqrt{3}aw/2$ is the ribbon width and $a$ is the graphene lattice constant. If $n = w + 1$ then $C/2 = \mathcal{W}$ which is equivalent to $N_{\text{t}} = 2N + 4$, where $N_{\text{t}}$ and $N$ are the number of atoms in the unit cell of the tube and ribbon, respectively. This decomposition also connects SWCNT(21, 21) with ZGNR(20).

The calculation for the nanotube with the diameter index $n$ gives the resonance energy for the nanoribbon with width index $w = n - 1$. These energies are the optical resonances predicted for an atlas of ZGNRs. Eq. (1) should provide a more precise estimate for ZGNRs with width index $w \geq 9$, since only the regular part of the linear mapping coefficients for $n \geq 10$ was used in their fitting (see Supplementary Fig. 7 in Supplementary Note 3). We have also plotted in Fig. 5 the green solid curves given by the Jiang et. al. fitting formula[49]. The formula provides the resonance energies of ZGNRs, which are close to the atlas obtained by mapping from the armchair SWCNTs. This is especially true for the first bulk resonance. In order to verify if this is indeed possible, we have performed similar calculations within the ZGNR (av.) TBM and have included these results to Fig. 5 as small open black circles. The black data points and green solid curves show scaling with the ribbon width that is deceivingly similar to that of actual absorption resonances given by large red open circles of the ZGNR atlas. However, the optical resonances allowed by selection rules must originate from the van Hove singularities that are asymmetrically rather than symmetrically situated with respect to the Fermi level. Therefore, we recommend to follow the atlas in order to avoid the mistakes in the optical resonance energy estimation. Based from the results presented above, we should mention that the linear mapping similar

to Eq. (1) should be possible between the armchair GNRs and zigzag SWCNTs due to their straightforward similarity in both electronic and optical properties[39–42].

## Discussion

In addition to the atlas application, the $2N + 4$-rule for the optical resonance alignment can be utilized for building functional carbon materials on a molecular scale and for designing multi-frequency resonant optoelectronic devices. It is also important to note that the geometrical match within the tube-ribbon pairs can be a source of undesired cross-talk between the different parts of complex optoelectronic circuits when the optical resonance line width is larger than the standard deviation $\sqrt{-\beta \ln(L_{\text{c}})}$ estimated from the AC. Such a cross-talk can manifest itself in the cloaking of the hybrid GNR@SWCNT systems[63,64], making one of the constituents invisible for optical characterization in a specific spectral range. Therefore, special care must be taken in the interpretation of measurements for such hybrid systems.

To conclude, we have revealed that the optical resonances of armchair single wall carbon nanotubes and zigzag graphene nanoribbons in parallel polarization of the incident light are related by a linear transformation when $n = w + 1$, where $n$ is the

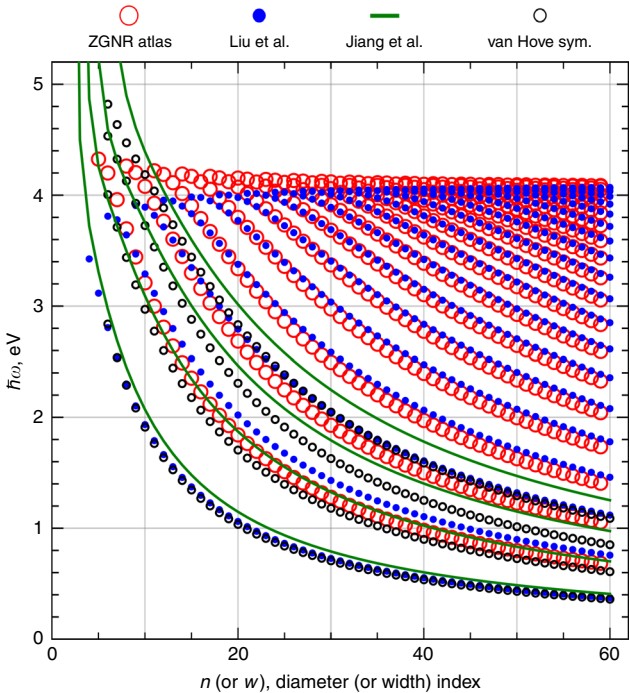

**Fig. 5 An atlas of ZGNR bulk optical resonances.** Optical resonances of the ZGNR atlas (large red open circles) obtained by linear mapping of the armchair SWCNT resonances (small blue filled circles) given by interpolating formula from Liu et al.[11] The fitting formula from Jiang et al.[49] (green solid curves) is added, as well as the energy differences (small open black circles) between the van Hove singularities symmetrically placed with respect to the Fermi level of ZGNRs (van Hove sym.) calculated based on ZGNR (av.) TBM from Table 1.

diameter(chiral) index and $w$ is the width index. The $n = w + 1$ relation is equivalent to the $2N + 4$-rule for the number of carbon atoms in a nanotube unit cell, where $N$ is the number of atoms in a ribbon unit cell. This rule is valid not only in the simple nearest-neighbor tight-binding model but also in more complicated models following the density functional theory calculations and simulating the electron exchange and correlation effects with several phenomenological parameters. Despite the different topology of the structures (cylindrical and planar), optical selection rules and notorious edge states, a high degree of alignment coefficient (>0.99) is found for the bulk-bulk transitions of ZGNR($w$) and for all but the first $\pi$-electron optical resonances of armchair SWCNT($n,n$), when the $2N + 4$-rule is fulfilled. This allows one to predict the bulk-bulk transitions of zigzag GNRs by mapping the atlas of armchair SWCNTs optical transitions of Liu et al.[11] Finally, we note that the atlas for nanoribbons can be further expanded to include the armchair GNR optical resonances, since similar mapping should be possible between the zigzag SWCNTs and armchair GNRs. Also, it would be interesting to investigate the effects of external electric and magnetic fields[65] which—as we anticipate—could be used to tune the degree of correlation and alignment between the optical absorption resonances.

## Methods

**DFT calculations.** All density functional theory (DFT) calculations were performed in Quantum Espresso package using the norm-conserving pseudopotential with Perdew-Burke-Ernzerhof exchange and correlation[66]. In these calculations, we used 40 Ry kinetic energy cutoff for the plane wave expansion and set the vacuum distances between periodic images of the structures to be about twice the transverse size of the structure (width for the ribbons and diameter for the tubes). In the majority of the band-structure calculations, the Brillouin zones were sampled with

$30 \times 1 \times 1$ uniform $k$-points mesh. For ZGNR(20) and SWCNT(21,21), we used a denser mesh of $60 \times 1 \times 1$ $k$-points and the kinetic energy cutoff of 100 Ry. For the sake of tight-binding model (TBM) parameters fitting, in all cases we used the non-relaxed ideal geometries of the structures (see Supplementary Discussion). The single-electron optical absorption spectra based on DFT wavefunctions were calculated within the pw2gw package of Quantum Espresso.

**Tight-binding calculations.** In the tight-binding calculations, we first followed an idealized picture where a single-parameter tight-binding model (TBM) was used for both tubes and ribbons by setting the nearest-neighbor hopping integral $t_1 = 3.12$ eV (Partoens 2006)[55]. Then, for zigzag GNRs, 6 TBM parameters were fitted to the DFT results to facilitate the study taking into account exchange and correlation effects for a wide range of transverse sizes. The fitted parameters are given in Table 1. For armchair SWCNTs, we used 7 tight-binding parameters reported by Reich et al. (Reich 2002)[60] because they perfectly describe the calculated DFT band-structures of the carbon nanotubes (see Suplementary Note 1). These parameters are also given in Table 1.

For all TBMs, the single-electron optical absorption spectra for the incident light polarized along the longitudinal axis of the nanoribbon or nanotube were calculated using the gradient approximation for the velocity operator[67]: $v = \partial H(k)/\partial k$. Then, with this operator the interband absorption is calculated as follows[27,56]:

$$A(\hbar\omega) \sim \sum_{i,j} \sum_{k} \frac{\left|\langle \Psi_{c,i}|v|\Psi_{v,j}\rangle\right|^2}{E_{c,i}(k) - E_{v,j}(k)} \delta(\hbar\omega - E_{c,i}(k) + E_{v,j}(k)), \quad (2)$$

where $\Psi_{c(v),i(j)}$ are the eigenvectors of the generalized eigenproblem $H\Psi = ES\Psi$ and $\delta(E)$ is the Dirac delta function that is replaced in numerical calculations by a Gaussian function with broadening $\alpha$ to model a range of de-excitation processes, i.e., $\delta(E) \rightarrow \exp\{-E^2/\alpha^2\}$. Eq. (2) assumes the zero temperature. The summation $\sum_k$ in Eq. (2) runs through the Brillouin zone sampled with 1000 $k$-points for the energy band-structure calculations. The broadening of a single transition was chosen to be $\alpha = 0.03$ eV. This broadening corresponds to relaxation time $\tau = 0.14$ ps that is a medial value for a range of the reported values for far-infrared and optical regimes[68]. The absorption spectrum in our calculations was sampled with 2000 points in the energy range from 0 to 8 eV, which corresponds to the resolution of about 8 points per the Gaussian.

**Fitting procedure.** Using Quantum Espresso package, we calculated the DFT band-structures for ZGNR(6), (9), (12) and (15). In each DFT band-structure, only the $\pi$-bands were selected for the fitting. The fitting was implemented on all valence $\pi$-bands and several lowest conductions $\pi$-bands: 6 valence and 2 conduction bands for ZGNR(6), 9 valence and 3 conduction bands for ZGNR(9), 12 valence and 4 conduction bands for ZGNR(12) and 15 valence and 5 conduction bands for ZGNR(15). The number of conduction bands was limited based on the pseudopotential vacuum level of about 4 eV; for fitting we chose only those conduction bands that lay below the vacuum level within the whole Brillouin zone. Each refined in this way DFT band-structure was fitted with six parameters of TBM. In order to fit the six parameters $\{p_l\}_{l=1,\dots,6} = \{t_1, t_2, t_3, s_1, s_2, s_3\}$, we minimized the following function:

$$f(\{p_l\}) = \sum_{i=1}^{N_k} \sum_{j=1}^{N_b} \left[E_{\text{DFT},j}(k_i) - \text{Re}\left(E_{\text{TBM},j}(k_i, p_l)\right)\right]^2 + \left[\text{Im}\left(E_{\text{TBM},j}(k_i, p_l)\right)\right]^2, \quad (3)$$

where $N_k$ is the number of the Brillouin zone sampling points used in DFT calculations and $N_b$ is the number of fitted bands. For the generalized eigenvalue problem $H\Psi = ES\Psi$, the Hermiticity of $H$ and $S$ matrices is not a sufficient condition for having real eigenvalues. All real eigenvalues are guaranteed for $H\Psi = ES\Psi$ problem only if $S$ and $H$ are not only Hermitian but also at least one of them is positive definite. This condition, however, is not fulfilled for many combinations of the hopping integrals varied between $-3$ and 3 eV and the overlapping integrals between $-1.5$ and 1.5. Therefore, to ensure realvaluedness of $f(\{p_l\})$ and to exclude the sets of the tight-binding parameters resulting in complex eigenenergies, the sum of the squares of the deviations between the DFT and TBM energies in the first term of Eq. (3) is placed within a parabolic well described by the second term in Eq. (3). The mimimizations were performed in Mathematica 11.3.0 software based on its built-in NMinimize function designed to find the global minimum. Among the six methods given for this function, only the differential evolution method showed a steady convergence within <200 steps, i.e., for ZGNR(6) $f(\{p_l\})_{\min} = 1.6780$ eV$^2$ within 133 steps, for ZGNR(9) $f(\{p_l\})_{\min} = 2.7957$ eV$^2$ within 110 steps, for ZGNR(12) $f(\{p_l\})_{\min} = 3.5849$ eV$^2$ within 155 steps and for ZGNR(15) $f(\{p_l\})_{\min} = 4.4504$ eV$^2$ within 186 steps. The resulting parameters presented in Table 1 of the main text were then averaged over the four structures and were used in the TBM calculations for the ZGNRs. Before TBM optical absorption calculations, we checked that the found parameters provide a reasonable approximation to the DFT band-structures (see Supplementary Figs. 1 and 2 in Supplementary Note 1).

**Absorption resonance extraction**. In each TBM absorption spectrum, the positions of the peaks were extracted using the algorithm (see Code Availability section) which identifies the positions of intensity maxima $E_{max}$ for which the minimum intensity drop $\Delta I_{min}$ at the left or right boundary of the closed interval $[E_{max} - dE, E_{max} + dE]$ fulfills the following criterion:

$$\frac{x}{100\%} \cdot I(E_{max}) < \Delta I_{min} \equiv I(E_{max}) - \max[I(E_{max} + dE), I(E_{max} - dE)] , \quad (4)$$

where $I(E)$ is the intensity of the interband absorption at a given energy and $x$ is the threshold parameter measured in terms of percents.

In order to extract the significant peaks from the full and bulk ZGNR spectra in Patroens 2006 TBM, we used the following parameters in Eq. (4): $x = 8\%$ and $dE = 0.07$ eV. To determine the peak positions for similar spectra calculated with ZGNR(av.) TBM presented in Table 1, we used $x = 4\%$ and $dE = 0.1$ eV for both full and bulk absorption spectra. In the case of armchair SWCNTs, the optical absorption resonances were selected by the algorithm with $x = 19\%$ and $dE = 0.1$ eV for Partoens 2006 and $x = 7\%$ and $dE = 0.05$ eV for Reich 2002 TBMs.

**Linear correlation coefficient**. The following modification of the standard formula [cf. Equation (7.18) in ref. [69]] for the linear correlation coefficient (LCC) calculation was used:

$$r = \frac{\sum_{i=1}^{\min(N_x, N_y)} (x_i - \bar{x})(y_i - \bar{y})}{\sqrt{\sum_{i=1}^{N_x} (x_i - \bar{x})^2 \sum_{i=1}^{N_y} (y_i - \bar{y})^2}}, \quad (5)$$

where $N_x$ and $N_y$ are the number of elements in datasets $x$ and $y$ representing the absorption resonances extracted from the spectra, while $\bar{x}$ and $\bar{y}$ are the mean values for the $x$ and $y$ datasets, respectively. In general, the numbers of resonances $N_x$ and $N_y$ in the absorption spectra of tubes and ribbons are different. Due to this reason, the summation in the numerator of Eq. (5) is defined for the less of $N_x$ and $N_y$. In this way, the LCC values were determined for all the combinations of tubes and ribbons of different transverse sizes and the LCC maps were generated.

**Alignment coefficient or the likelihood function**. The degree of alignment $L_c$ between the two sets of resonances was described by the following alignment coefficient (AC) or the likelihood function, characterizing the average deviation of $x_i$ resonances from $y_i$ ones (cf. Equation (6.2) in ref. [69]):

$$L_c = \exp\left[ -\frac{\sum_{i=1}^{\min(N_x, N_y)} (x_i - y_i)^2}{\beta \cdot \min(N_x, N_y)} \right], \quad (6)$$

where $\beta = 1$ eV$^2$ is a coefficient setting the energy units of $x_i$ and $y_i$ resonances and making the exponent dimensionless. Similar to the LCC, the AC defined by Eq. (6) truncates the dataset with a larger number of elements to the length of the smaller dataset, which allows one to generate the AC maps.

## Data availability

The TBM and DFT calculations raw data that support the findings of this study are available in Zenodo with the identifier doi:10.5281/zenodo.3547546 or using link https://doi.org/10.5281/zenodo.3547546. This open access repository also contains Source Data files for Fig. 1 and Table 1. In addition, the repository contains scripts for performing the TBM calculations, TBM fitting and data analysis, including reproduction of Figs. 1–5 and Supplementary Figs. 1–8. These data and code can be alternatively obtained from the GitHub (see Code availability section). The data that support the findings of this study are also available from the corresponding author upon request.

## Code availability

The code developed for this study is available under a GNU Lesser General Public License v3.0 from https://github.com/vasilsaroka/An-atlas-of-ZGNRs-bulk-optical-resonances.

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

## Acknowledgements

The authors thank C.A. Downing, R.R. Hartmann, M.E. Portnoi, K.G. Batrakov, P.P. Kuzhir, A.L. Pushkarchuk, A.V. Kuhta, F. Yao and K. Liu for their very helpful discussions, J. Danon for providing computational facilities at NTNU and A.R. Villagracia for providing the computing workstation in STRC, De La Salle University. CPU time for the DFT calculations was granted by CINECA HPC center. This work was supported by EU H2020 RISE Project CoExAN (GA644076), and partly by the Research Council of Norway Center of Excellence funding scheme (project no. 262633, "QuSpin"). The authors also acknowledge the financial support from VCRI, De La Salle University.

## Author contributions

V.A.S. and D.I.L. conceived the idea. R.B.P. obtained the first DFT results. R.B.P., G.N.C.S., D.G. and O.P. performed DFT calculations on small and large structures and analysed the results. V.A.S. performed the fitting of TB models and TB calculations. D.I.L. and V.A.S. compared the results to the interpolation formulas for carbon nanotubes. V.A.S. and R.B.P. wrote the manuscript. All authors commented on the manuscript and assisted in revisions.

## Competing interests

The authors declare no competing interests.
