## [Peer Review File · Nature Communications]

Reviewers' comments:

Reviewer #1 (Remarks to the Author):

[[Decision]]: minor revision

[[Comment]]:

Since Kataura published his work on [Synth. Met. 103, 2555 (1999)], the famous Kataura plot is well known by scientific researchers for providing the relations among chirality, diameter, and optical transitions of carbon nanotube (CNT). A Kataura plot for the quasi-one-dimensional graphene nanoribbons (GNRs) is also important and required in the research field. GNRs can be treated as unrolled CNTs, it seems that the ribbon version Kataura plot can be obtained easily. However, as the authors told: "it is not easily applicable because of two main reasons, namely, (i) the different physical mechanisms of SWCNT and GNR synthesis and (ii) a huge variety of GNR edges which impedes the development of their standard classification due to different naming conventions used in experimental and theoretical studies."

In this work, the authors have presented the incident light optical resonances in armchair single-wall carbon nanotubes and zigzag graphene nanoribbons, which are related by a linear transformation if between the tube diameter (chiral) index and the ribbon width index. Thousands of optical spectra are analyzed (It's hard work). Hence, a "2N+4-rule" is found and an atlas of bulk optical resonances of zigzag graphene nanoribbon (a ribbon version Kataura plot) is exhibited.

The exchange and correlation effects are also considered in the theoretical model. Some fundamental results from other previous works are also noted. The validity of the rule is not only in the nearest-neighbor tight-binding model but also in more complicated models with several phenomenological parameters simulating the electron exchange and correlation effects following the density functional theory calculations.

In concluding the aforementioned points, this article could be published after minor revision. Some suggestions are listed below.

[[Suggestions]]

[Abstract]

The abstract should give the readers conceptual and concise information about the work. Some description given too many details should be omitted. In this abstract, too many numbers are provided (e.g. 693 optical absorption resonances of 57 armchair carbon nanotubes and 1041 absorption resonances of 59 zigzag graphene nanoribbons continuously spanning the tube (ribbon) transverse sizes from 0:5(0:4) nm to 8:1(12:8) nm). We know that the authors want to show their hard work and efforts by large numbers. This is an appropriate treatment. However, don't do this too much. Just once, it's sufficient, and the readers will have a strong impression on the article.

For example, in the article [Nat. Nanotechnol. 7, 325 (2012)], only a number "206" in the abstract, and the readers do know the authors hard work.

"Here, we report the establishment of a structure-property 'atlas' for nanotube optical transitions based on simultaneous electron diffraction measurements of the chiral index and Rayleigh scattering measurements of the optical resonances of 206 different single-walled nanotube structures."

[Result]

Please use subsection structure in the result section. The subsection structure is a well composing technique for presenting complex research work in a step by step way. So, choose a suitable title

for subsection is very important. The readers will understand what's next in the subsection. Also, it's a good way to avoid a paragraph starting from "In figure X,"

[Figure]

The authors give suitable figure titles for figure 1 to 4. Brief information about the figure is given, e.g., Comparison between DFT (solid gray) and fitted TBM (dashed black) optical absorption spectra of SWCNT(21; 21) (tube) and ZGNR(20) (ribbon). However, the title of figure 5 is "Kataura plot." It's too short, and readers might guess the authors paste the original Kataura plot in [Synth. Met. 103 (1999) 2555].

[Method]

The research method is described in many places in the result section. These statements of the method are suggested to move to the "Method" section after "Discussion." Use this article structure, the readers won't be bothered by the detail method descriptions and will focus on the data analysis in the result section.

[Optical transition under external fields]

The major results present the regularity of optical transitions under no external fields. In addition, the effect of external fields could be considered to expand the study, e.g. the effect of Landau quantization for the optical transition [Phys. Chem. Chem. Phys. 18, 7573 (2016)]. To do whole research work on a further topic might take a lot of time and effort. To discuss the further topic and give the readers some physical picture is a much better way. I wish the article can give scientific researchers not only the rigorous research results but also more imagination.

Reviewer #2 (Remarks to the Author):

In this manuscript, B. Payod et al. revealed a high degree of correlation (' $2N+4$ ' rule) between the optical absorption resonances of armchair carbon nanotubes and zigzag graphene nanoribbons, which was supported by high linear correlation and alignment coefficient mathematically. Based on this rule and the atlas of carbon nanotubes optical transitions [Nat. Nanotechnol.7, 325, 2012], atlas of optical resonances of zigzag graphene nanoribbons can be obtained. I think this work is enlightening and provide a basis for future research on one dimensional nanoribbons. The theory itself is very solid and convincing. The only concern is that whether this graphene nanoribbon atlas can be really applied as that in carbon nanotubes. In the below I have some specific comments on the content.

(1) As the authors claimed in this paper, multi parameter tight-binding model (TBM), which incorporated the many-body effect such as Coulomb electron-electron interaction, can give a more realistic picture. However, in Figure 2, it seems that results calculated by single-parameter TBM have a better fit to the theoretical reference line ($n = w + 1$). In addition, the edge states affect the optical resonances of nanoribbon a lot, but this perfect agreement with the reference line is obtained exclusion of the edge states. Basically, the many-body effects and edge states effects should be considered seriously.

(2) For clarity, an intensity bar, or a note at least, can be given in LCC maps and the following figures.

(3) Since simulated optical resonances of carbon nanotube have been measured and reproduced experimentally. If possible, can the author confirm the calculated atlas of nano ribbon by providing a proper comparison with existed experiment results?

(4) The interrelations between the optical absorption resonances of armchair carbon nanotubes and zigzag graphene nanoribbons is proven mathematically. Can authors give the deeper physical analysis and understanding of the correlation? Or give more insight into connection of CNT with other nanoribbons with different structures?

Point-to-point Response:

Reviewer #1:

Since Kataura published his work on [Synth. Met. 103, 2555 (1999)], the famous Kataura plot is well known by scientific researchers for providing the relations among chirality, diameter, and optical transitions of carbon nanotube (CNT). A Kataura plot for the quasi-one-dimensional graphene nanoribbons (GNRs) is also important and required in the research field. GNRs can be treated as unrolled CNTs, it seems that the ribbon version Kataura plot can be obtained easily. However, as the authors told: “it is not easily applicable because of two main reasons, namely, (i) the different physical mechanisms of SWCNT and GNR synthesis and (ii)

a huge variety of GNR edges which impedes the development of their standard classification due to different naming conventions used in experimental and theoretical studies.”

In this work, the authors have presented the incident light optical resonances in armchair single-wall carbon nanotubes and zigzag graphene nanoribbons, which are related by a linear transformation if between the tube diameter (chiral) index and the ribbon width index. Thousands of optical spectra are analyzed (It’s hard work). Hence, a “ $2N+4$ -rule” is found and an atlas of bulk optical resonances of zigzag graphene nanoribbon (a ribbon version Kataura plot) is exhibited.

The exchange and correlation effects are also considered in the theoretical model. Some fundamental results from other previous works are also noted. The validity of the rule is not only in the nearest-neighbor tight-binding model but also in more complicated models with several phenomenological parameters simulating the electron exchange and correlation effects following the density functional theory calculations.

In concluding the aforementioned points, this article could be published after minor revision. Some suggestions are listed below.

Response: We appreciate deep and thorough analysis of our results and a strong support of our work by the Reviewer.

Comment 1. [Abstract]

The abstract should give the readers conceptual and concise information about the work. Some description given too many details should be omitted. In this abstract, too many numbers are provided (e.g. 693 optical absorption resonances of 57 armchair carbon nanotubes and 1041 absorption resonances of 59 zigzag graphene nanoribbons continuously spanning the tube (ribbon) transverse sizes from 0:5(0:4) nm to 8:1(12:8) nm). We know that the authors want to show their hard work and efforts by large numbers. This is an appropriate treatment. However, don’t do this too much. Just once, it’s sufficient, and the readers will have a strong impression on the article.

For example, in the article [Nat. Nanotechnol. 7, 325 (2012)], only a number “206” in the abstract, and the readers do know the authors hard work.

“Here, we report the establishment of a structure-property ‘atlas’ for nanotube optical transitions based on simultaneous electron diffraction measurements of the chiral index and Rayleigh scattering measurements of the optical resonances of 206 different single-walled nanotube structures.”

Response 1. This is a very good advice. Following the Reviewer suggestion, we have omitted the number of studied structures and optical absorption resonances while keeping the range of transverse sizes of armchair carbon nanotubes and the zigzag graphene nanoribbons. This change can give the readers a good impression to the breadth of analysis done in this work without dealing into specifics.

The original abstract was

“By combining ab-initio Density Functional Theory with Tight-Binding calculations, we investigate 693 optical absorption resonances of 57 armchair carbon nanotubes and 1041 absorption resonances of 59 zigzag graphene nanoribbons continuously spanning the tube (ribbon) transverse sizes from 0.5(0.4) nm to 8.1 (12.8) nm. Analysis of the transverse size dependence for both structure types shows that all but the lowest in energy π -electron optical resonances in armchair carbon nanotubes linearly map to the bulk-bulk resonances in zigzag graphene nanoribbons. We reveal a ‘ $2N+4$ ’-rule for the design of carbon-based optoelectronic nanodevices and nanocircuits by showing that the linear mapping is guaranteed only when the number of atoms in the tube unit cell is $2N+4$, where N is the number of atoms in the ribbon unit cell. Using this rule, we map the atlas of carbon nanotubes optical transitions [K. Liu et. al., Nat. Nanotechnol. 7, 325 (2012)] to an atlas of optical resonances of zigzag graphene nanoribbons. The atlas represents an important tool for fast optical characterization of the nanoribbons, paving the way towards reliable spectroscopy of carbon based nanodevices.”

The updated abstract is

“By combining the ab-initio Density Functional Theory with Tight-Binding calculations, we investigate the optical absorption resonances of armchair carbon nanotubes and zigzag graphene nanoribbons continuously spanning the tube (ribbon) transverse sizes between 0.5(0.4) nm to 8.1(12.8) nm. Analysis of the transverse size dependence for both structures shows that optical resonances of the armchair carbon nanotubes (excluding the lowest energy π -electron resonances) linearly map to the bulk-bulk resonances of zigzag graphene nanoribbons. We reveal a ‘ $2N+4$ ’-rule for the design of carbon-based optoelectronic nanodevices and nanocircuits by showing that the linear mapping is guaranteed when the number of atoms in the tube unit cell is $2N+4$, where N is the number of atoms in the ribbon unit cell. Applying this rule, we show that an atlas of carbon nanotubes optical transitions can be mapped to an atlas of optical resonances of zigzag graphene nanoribbons. This represents an important tool for fast optical characterization of graphene nanoribbons, paving the way towards reliable spectroscopy of carbon based nanodevices.”

Comment 2. [Result]

Please use subsection structure in the result section. The subsection structure is a well composing technique for presenting complex research work in a step by step way. So, choose a suitable title for subsection is very important. The readers will understand what’s next in the subsection. Also, it’s a good way to avoid a paragraph starting from “In figure X,”

Response 2. This advice is truly helpful for keeping the manuscript well-organized. We have divided the Results section into 5 subsections:

1. Optical absorption of armchair SWCNTs and zigzag GNRs.
2. Linear correlation coefficient analysis.

3. Alignment coefficient analysis.
4. '2N+4'-rule of armchair SWCNTs and ZGNRs
5. An atlas of ZGNR absorption resonances by linear mapping

The divisions are based on the scheme used for presenting the data and how they add up to the conclusion of the 2N+4-rule between the armchair SWCNT and ZGNR absorption resonances.

Comment 3. [Figure]

The authors give suitable figure titles for figure 1 to 4. Brief information about the figure is given, e.g., Comparison between DFT (solid gray) and fitted TBM (dashed black) optical absorption spectra of SWCNT(21; 21) (tube) and ZGNR(20) (ribbon). However, the title of figure 5 is "Kataura plot." It's too short, and readers might guess the authors paste the original Kataura plot in [Synth. Met. 103 (1999) 2555].

Response 3. We thank the Reviewer 1 for this comment. We have changed the caption to clearly describe the plot.

The original caption of Figure 5 was

"Kataura plots: (open red circles) ZGNRs optical resonances, obtained by linear mapping of Liu et al. interpolating formula for armchair SWCNTs (filled blue circles),²² (green solid curves) fitting formula from ref 73; (open black circles) energy differences between the conduction subband minima and valence subband maxima which correspond to the van Hove singularities symmetrically placed with respect to the Fermi level of ZGNRs [same scheme as used in ref 73 but implemented within the ZGNR (av.) TBM from Table 1]."

The updated caption of Figure 5 is

"A comparison between the Kataura-type plots for ZGNRs. Optical resonances for different ribbon widths (open red circles) are obtained by linear mapping of the interpolating formula for armchair SWCNTs from Liu et al.²² (filled blue circles). The fitting formula from Jiang et al.⁹⁰ (green solid curves) is compared with the energy differences between the conduction subband minima and valence subband maxima calculated based on 'ZGNR (av.)' TBM from Table 1 (open black circles). The later energy calculations correspond to the van Hove singularities symmetrically placed with respect to the Fermi level of ZGNRs."

Comment 4. [Method]

The research method is described in many places in the result section. These statements of the method are suggested to move to the "Method" section after "Discussion." Use this article structure, the readers won't be bothered by the detail method descriptions and will focus on the data analysis in the result section.

Response 4. This suggestion is really practical for keeping the flow of discussion on the results smooth and informative. We have moved the methods to the 'Methods' section at the end of the manuscript. The details of the methods are presented in 3 subsections:

1. Tight-binding and DFT calculations.
2. Linear correlation coefficient.
3. Alignment coefficient or the 'likelihood function'.

Comment 5. [Optical transition under external fields]

The major results present the regularity of optical transitions under no external fields. In addition, the effect of external fields could be considered to expand the study, e.g. the effect of Landau quantization for the optical transition [Phys. Chem. Chem. Phys. 18, 7573 (2016)]. To do whole research work on a further topic might take a lot of time and effort. To discuss the further topic and give the readers some physical picture is a much better way. I wish the article can give scientific researchers not only the rigorous research results but also more imagination.

Response 5. The Reviewer is right. The inclusion of the external fields into consideration is one of the possible future developments of the present study. From our point of view, the most interesting aspect of such a study could be the scrutinizing of the tunability of the degree of the alignment and linear correlation. This could be useful, in particular, for the development of multi-resonance switching devices with possible applications in quantum information. In view of this we have outlined the future development for an interested reader:

“Finally, we note that the atlas for nanoribbons can be further expanded to include the armchair GNR optical resonances, since similar mapping must be possible between the zigzag SWCNTs and armchair GNRs. Also, it would be interesting to investigate the effects of external electric and magnetic fields¹¹⁰ which -- as we anticipate -- could be used to tune the degree of correlation and alignment between the optical absorption resonances.”

=====

Reviewer #2:

In this manuscript, B. Payod et al. revealed a high degree of correlation ('2N+4' rule) between the optical absorption resonances of armchair carbon nanotubes and zigzag graphene nanoribbons, which was supported by high linear correlation and alignment coefficient mathematically. Based on this rule and the atlas of carbon nanotubes optical transitions [Nat. Nanotechnol.7, 325, 2012], atlas of optical resonances of zigzag graphene nanoribbons can be obtained. I think this work is enlightening and provide a basis for future research on one dimensional nanoribbons. The theory itself is very solid and convincing. The only concern is that whether this graphene nanoribbon atlas can be really applied as that in carbon nanotubes. In the below I have some specific comments on the content.

Response: We thank the Referee for the support and high estimate of our work. The atlas is indeed needed for the fast and non-invasive ribbon width characterization especially in those methods that provide flexible control of the ribbon width such as catalytic writing [1],

epitaxial grown on sidewalls of SiC [2], nanolithography [3], plasma etching[4] or tube unzipping [5].

- [1] L. C. Campos, V. R. Manfrinato, J. D. Sanchez-Yamagishi, J. Kong, and P. Jarillo-Herrero, *Nano Lett.* **9**, 2600 (2009); X. Cao, Y. Ji, W. Hu, S. Duan, and Y. Luo, *J. Phys. Chem. C* **118**, 22643 (2014).
- [2] I. Palacio, A. Celis, M. N. Nair, A. Gloter, A. Zobelli, M. Sicot, D. Malterre, M. S. Nevius, W. A. de Heer, C. Berger, E. H. Conrad, A. Taleb-Ibrahimi, and A. Tejada, *Nano Lett.* **15**, 182 (2015); M. S. Nevius, F. Wang, C. Mathieu, N. Barrett, A. Sala, T. O. Montes, A. Locatelli, and E. H. Conrad, *Nano Lett.* **14**, 6080 (2014); J. Baringhaus, J. Aprojanz, J. Wiegand, D. Laube, M. Halbauer, J. Hübner, M. Oestreich, and C. Tegenkamp, *Appl. Phys. Lett.* **106**, (2015).
- [3] L. Tapasztó, G. Dobrik, P. Lambin, and L. P. Biró, *Nat. Nanotechnol.* **3**, 397 (2008); G. Z. Magda, X. Jin, I. Hagymási, P. Vancsó, Z. Osváth, P. Nemes-Incze, C. Hwang, L. P. Biró, and L. Tapasztó, *Nature* **514**, 608 (2014).
- [4] G.-L. Wang, L. Xie, P. Chen, R. Yang, D.-X. Shi, and G.-Y. Zhang, *Acta Phys. Sin.* **65**, 196101 (2016); G. Wang, S. Wu, T. Zhang, P. Chen, X. Lu, S. Wang, D. Wang, K. Watanabe, T. Taniguchi, D. Shi, R. Yang, and G. Zhang, *Appl. Phys. Lett.* **109**, (2016); X. Wang and H. Dai, *Nat. Chem.* **2**, 661 (2010); X. Zhang, O. V. Yazyev, J. Feng, L. Xie, C. Tao, Y.-C. Chen, L. Jiao, Z. Pedramrazi, A. Zettl, S. G. Louie, H. Dai, and M. F. Crommie, *ACS Nano* **7**, 198 (2013).
- [5] E. Cunha, M. F. Proença, F. Costa, A. J. Fernandes, M. A. C. Ferro, P. E. Lopes, M. González-Debs, M. Melle-Franco, F. L. Deepak, and M. C. Paiva, *ChemistryOpen* **4**, 115 (2015); L. Jiao, L. Zhang, X. Wang, G. Diankov, and H. Dai, *Nature* **458**, 877 (2009); D. V Kosynkin, A. L. Higginbotham, A. Sinitskii, J. R. Lomeda, A. Dimiev, B. K. Price, and J. M. Tour, *Nature* **458**, 872 (2009); S. Y. Bang, J. H. Ryu, B. G. Choi, and K. B. Shim, *J. Alloys Compd.* **618**, 33 (2014).

Following the Reviewer's comment we have updated the following paragraph of the manuscript:

“A standard way of presenting diameter dependence of SWCNT optical resonances polarized parallel to the tube axis is Kataura plot.¹⁵ This plot has been both measured experimentally^{16,17,19,28} and reproduced numerically in semi-empirical and ab-initio calculations.^{29,31} Several interpolating formulas have been proposed for the description of the SWCNT Kataura plot.^{21,23} The Kataura plot for ribbons cannot be approached in the same way as Kataura plot for tubes because of (i) the different physical mechanisms of SWCNT and GNR synthesis and (ii) a huge variety of GNR edges and different naming conventions used in experimental and theoretical studies which impedes the development of the GNR standard classification.^{32,43} Limiting our consideration to the symmetric nanoribbons with zigzag and armchair edges, we should notice that even these two types of GNRs are not yet well studied and understood. In particular, no Kataura plot has been reported for these graphene nanoribbons, although such plot would be practical for the fast and non-invasive ribbon width characterization, especially in those synthesis methods that provide flexible control of the ribbon width, such as catalytic writing,^{44,45} epitaxial growth on sidewalls of SiC,⁴⁶⁻⁴⁸ nanolithography⁴⁹⁻⁵¹ plasma etching⁵²⁻⁵⁵ or tube unzipping⁵⁶⁻⁵⁹ (see also Ref. 60)”

Comment 1. As the authors claimed in this paper, multi parameter tight-binding model (TBM), which incorporated the many-body effect such as Coulomb electron-electron interaction, can give a more realistic picture. However, in Figure 2, it seems that results calculated by single-parameter TBM have a better fit to the theoretical reference line ($n = w + 1$). In addition, the edge states affect the optical resonances of nanoribbon a lot, but this perfect agreement with the reference line is obtained exclusion of the edge states. Basically, the many-body effects and edge states effects should be considered seriously.

Response 1. We thank the Reviewer for noticing this place in the manuscript that uses confusing wording ‘realistic’. We must say that we appreciate both the simple tractable model like the nearest neighbor TBM and more complicated semi-empirical and first principle models. We use the full range of models to investigate the reported effect and we show that it can be easily identified in all the models.

The behavior presented in Fig. 2 is correct one. In single parameter model that does not incorporate many-body effects the only difference between the tubes and ribbons can be related (1) to the structure of their lattices, (2) to the boundary conditions for the electron wave function leading to the quantization of electron momentum. With respect to the former we notice that both structures are based on graphene hexagonal lattice with the same crystallographic orientation of the vector of the translation invariance. As for the latter, the quantization of the electron momentum is approximately the same for the structures when “ $2N+4$ ”-rule is fulfilled (more details in Response to Comment 4). Therefore, it is expected to have higher degree of correlation in single parameter models when the same value of the hopping integral (for instance obtained by fitting data for graphene) is used for tubes and ribbons.

The main objective of Figure 2 is to show our readers the whole picture of the transverse size dependence. The many body effects are treated with a great precision in the DFT calculations presented in Figure 4 which corroborates a good agreement with the first principle calculations. We should note that for now fully first principle calculations of the LCC maps is beyond the abilities of any research group in the world.

We should clarify that the edge states themselves do not affect positions of the optical resonances originating from the bulk-bulk optical transitions. This is clearly seen from the SI Figs. S3 and S4 in both tight-binding models with and without many-body effects. We report the correlation between bulk optical resonances. The data with edge states show a sensitivity of the linear correlation coefficient to the presence of uncorrelated peaks in the sets of data rather than the importance of the edge states. As follows from Figure 2 linear correlation coefficient is indeed quite sensitive and can be used as a reliable measure.

Following the Reviewer’s comment we have added the following clarifications to the manuscript:

- (1) “Then, several TBM parameters are fitted to *the results from Density Functional Theory (DFT) to facilitate the study taking into account exchange and correlation*

effects for the wide range of transverse sizes (for more details see Section A in Supporting Information (SI)).”

- (2) *“The type (i) and (ii) data sets cannot be obtained by analysis of the full absorption spectra of ZGNRs with the peak selection algorithm because it is not selective with respect to the edge-to-bulk or bulk-bulk nature of the optical resonances (see Section B in SI). An intelligent approach to isolating such data sets is to apply the peak selection algorithm to the bulk absorption spectra obtained by exclusion of the edge states from the summation in the optical absorption (see eq. S2 in Section B of SI). As one can clearly see from Figures S3 and S4 the inclusion or exclusion of the edges states from the summation does not affect the energies of the bulk-bulk resonances therefore this procedure is equivalent to the selection of the bulk absorption resonances from the full absorption spectra.”*
- (3) We have also added bulk spectra of ZGNR(20) to Figure 4 to support our point and facilitate comparison between full and bulk ZGNRs spectra:

In Figure 4 caption: *“The optical absorption $A(\hbar\omega)$ of SWCNT(21,21) and ZGNR(20) in the DFT (light gray) and fitted TBMs (dashed black). The bulk optical absorption of ZGNR(20) in the fitted TBM is depicted with light red in both SWCNT(21,21) and ZGNR(20) panels.”*

In the text: *“It can be seen in Figure 4 that indeed the DFT and fitted TBM absorption spectra are in full agreement. It is also seen in Figure 4 that bulk spectra of ZGNRs can be obtained from full spectra by exclusion of the edge states without affecting positions of the bulk resonances.”*

Comment 2. For clarity, an intensity bar, or a note at least, can be given in LCC maps and the following figures.

Response 2. We thank the Referee for this comment. Following the Referee’s advice, we have added notes explaining the usage of the color gradient and notation in the LCC and AC maps.

We have updated the captions of Figures 2 and 3 as follows:

Figure 2 caption:

“The linear correlation coefficient (LCC) maps for optical resonances of armchair SWCNTs (all but the lowest energy one) and ZGNRs: (a)-(f) single parameter Partoens2006 TBM; (g)-(l) fitted TBMs from Table 1. (a)-(c) and (g)-(i) LCC maps include the edge-to-bulk state transitions in ZGNRs. (d)-(f) and (j)-(l) LCC maps include only bulk-bulk state transitions in ZGNRs. Insets show full indiscriminate LCC maps, where the color gradient from black to white corresponds to the ascending LCC values. The red regions signify $r > r_{th}$, where r_{th} is the threshold value for the linear correlation coefficient. Red semitransparent lines are the reference lines given by $n = w + 1$.”

Figure 3 caption:

*“The alignment coefficient (AC) map for **the** optical absorption resonances in armchair SWCNTs (excluding the first resonance) and **the** bulk-bulk optical resonances in ZGNRs with (a)-(d) nearest-neighbor TBM $t_1=3.12\text{eV}^{105}$ and (e)-(h) fitted TBMs from Table 1. The red line is the reference line $n = w + 1$. Insets of the graphs displays the full AC map before discrimination, where the color gradient from black to white corresponds to the increase of the AC values. The dark red regions signify $L_c > L_{c,th}$, where $L_{c,th}$ is a chosen threshold for the alignment coefficient.”*

Comment 3. Since simulated optical resonances of carbon nanotube have been measured and reproduced experimentally. If possible, can the author confirm the calculated atlas of nano ribbon by providing a proper comparison with existed experiment results?

Response 3. For the moment, we are not aware of the proper data for comparison. However, from private correspondence with several experimental groups working on atomically precise ribbons [Prof. Roman Fasel (Empa Materials Science and Technology), Prof. Alexander Sinitskii (University of Nebraska-Lincoln), Prof. Michael Crommie (UC Berkeley)], we know that they are working on the techniques for transferring ribbons to the isolating substrate and measuring their optical properties.

Comment 4. The interrelations between the optical absorption resonances of armchair carbon nanotubes and zigzag graphene nanoribbons is proven mathematically. Can authors give the deeper physical analysis and understanding of the correlation? Or give more insight into connection of CNT with other nanoribbons with different structures?

Response 4. The physical interpretation is possible in the nearest neighbor tight-binding model. In this model the physical picture is the following. The linear correlation and alignment between the bulk-bulk transitions is related to the matching of the quantized transverse electron momenta in tubes and ribbons. Similar analysis within the **kp** model is in the progress.

We have added the following clarification to the manuscript:

“In brief, such matching is achieved by equating the transverse momenta θ quantized in the two nanostructures by the secular equations: $\cos(n\theta/2) = 1$ (for tube) and $\sin(w\theta) + 2 \cos(k/2) \sin[(w + 1)\theta] = 0$ (for ribbon) [note that ZGNR secular equation depends on the electron longitudinal momentum k]. By setting $k \rightarrow 0$, which will correspond to the approximate alignment of the ribbon energy bands to the tube ones at the center of the Brillouin zone,⁹⁶ the secular equation for the ribbon has solution $\theta_j^{(r)} = \pi j / (w + 1)$, while the solution for the tube is $\theta_j^{(t)} = \pi j / n$. The reference line is derived by setting $\theta_j^{(r)} = \theta_j^{(t)}$.”

We have also updated the inset of Figure 4 and added the following explanation on how the matching of the electron momenta relates to the structures of the tubes and ribbons in question:

“The inset in the top right corner shows the decomposition of SWCNT(7,7) with circumference C into two ZGNR(6) (red and green) with effective width \mathcal{W} , which accounts for the two zigzag chains of carbon atoms to be removed (blue). The effective ribbon width

$\mathcal{W} = W + \sqrt{3} a/2$, where $W = \sqrt{3} a w/2$ is the ribbon width and a is the graphene lattice constant. If $n = w + 1$ then $C/2 = \mathcal{W}$ which is equivalent to $N_t = 2N + 4$, where N_t and N are the number of atoms in the unit cell of the tube and ribbon, respectively. This decomposition also relates SWCNT(21,21) to ZGNR(20).”

REVIEWERS' COMMENTS:

Reviewer #2 (Remarks to the Author):

The authors have added more details of the used methodology and improved the writing of the article, which make the application of this calculation more clear. I can now support the paper publication in nature communications.

REVIEWERS' COMMENTS:

Reviewer #2 (Remarks to the Author): The authors have added more details of the used methodology and improved the writing of the article, which make the application of this calculation more clear. I can now support the paper publication in nature communications.

Response: We thank Reviewer #2 for his/her support of our work.